# Contextual Experience Replay for Continual Learning of Language Agents

## Abstract

Large language model (LLM) agents have been applied to sequential decision-making tasks such as web navigation, but without any environment-specific experiences, they often fail in these complex tasks. Moreover, current LLM agents are not designed to continually learn from past experiences during inference time, which could be crucial for them to gain these environment-specific experiences. To address this, we propose Contextual Experience Replay (CER), a training-free framework to enable efficient continual learning for language agents in their context window. Specifically, CER accumulates and synthesizes past experiences into a dynamic memory buffer. These experiences encompass environment dynamics and common decision-making patterns, allowing the agents to retrieve and augment themselves with relevant knowledge in new tasks, enhancing their adaptability in complex environments. We evaluate CER on the challenging WebArena and VisualWebArena benchmarks. On VisualWebArena, CER surpasses the tree search method with much fewer token costs and achieves state-of-the-art performance of 31.9%. On WebArena, CER also gets a competitive average success rate of 33.16%, relatively improving the success rate of the GPT-4o agent baseline by 36.6%. We also show that CER can work even better if provided with a few annotated trajectories or combined with other methods, demonstrating its potential.

## 1 Introduction

Building an autonomous agent that can help with people's daily tasks has been a long-standing goal of artificial intelligence research (Russell & Norvig, 1995; Franklin & Graesser, 1996). Recently, large language models (Achiam et al., 2023; Anthropic, 2024; Gemini Team, 2023) have shown impressive performance in text (Hendrycks et al., 2021) and code generation (Chen et al., 2021; Xie et al., 2024b), reasoning (Wei et al., 2022; Yao et al., 2023a), and decision-making tasks (Yao et al., 2023b; Zhou et al., 2024a; Xu et al., 2023; Xie et al., 2024c), which paves the way for building an agent to automate computer tasks. Web tasks, specifically, are a representative task type in computer tasks, which is more controllable than the OS environment (Xie et al., 2024a) and more complex than the mobile environment (Rawles et al., 2023; 2024). On two realistic web navigation benchmarks, WebArena (Zhou et al., 2024b) and VisualWebArena (Koh et al., 2024a), humans can achieve success rates of 78.24% and 88.70%, correspondingly. However, the current methods, with the most frontier models, can only achieve a success rate of around or less 20% without human involvement.

One important reason is the lack of prior knowledge of each environment, which is critical for such difficult multi-step task solving in the complex web environment. While training in each specific environment is costly, current language agents seldom have an efficient way to continually learn about the environment, so they need to explore the environment from scratch for every single task (Koh et al., 2024b).

In this work, we propose Contextual Experience Replay (CER), a novel and effective framework to enable the continual learning of language agents in complex environments. CER is loosely inspired by experience replay (Schaul et al., 2016; Rolnick et al., 2019), an important algorithm in reinforcement learning which highlights storing past trajectories into a buffer and training the agent with these data.

Figure 1: Overview of Contextual Experience Replay including offline and online settings. (1) In the online setting, it will start from stage C and loop between stage C and B for each task, i.e. solve task $i$, learn experiences from it and solve task $i + 1$ with previous experiences, and so on. (2) In the offline setting, stage A is needed to get offline trajectories, then it goes from stage B to C and finally stays in stage C, i.e., learns experiences from offline trajectories and solves all tasks. (3) In the hybrid setting, it will begin from stage A and loop between B and C, conducting both offline and online learning.

Our approach allows agents to distill experience from trajectories, including environment dynamics and common decision-making patterns, from past trajectories, store them into a dynamic memory, retrieve them with the current task, and replay them in context when solving new tasks. Fig.1 shows how CER works under different settings. Online, offline, and hybrid settings are divided by the source of trajectories, i.e., the time to get the trajectories. As in Fig.1, in the online setting, the agent will start from the inference stage (C) without any experience. After completing a task, CER gets the (online) trajectory from it, distills experiences from the trajectories, and merges it into the buffer. During the inference of the next task, the agent will be augmented with retrieved helpful experiences and so on. In the offline setting, a set of trajectories will be collected in advance (stage A), distilled into experiences, and stored. Then, the agent will solve tasks on the test set with retrieved experience from the fixed buffer. The hybrid setting is the combination of these two, i.e., going through the offline learning stage before online learning. Fig. 2 also shows how the experience is utilized by the agent with an example.

We evaluated CER on two realistic web benchmarks WEBARENA (Zhou et al., 2024b) and VISUAL-WEBARENA (Koh et al., 2024a). CER improves the GPT-4o baseline by a large margin and achieves competitive results on these two benchmarks while orthogonal with most other methods. On WE-BARENA, CER shows a relative improvement of 33.7% over the GPT-4o baseline and achieves an overall success rate of 33.2%, competitive with other state-of-the-art (SOTA) methods. On VISU-ALWEBARENA, CER outperforms the tree search-based method by 20.8% in relative performance with dozens of times fewer token costs and achieves a SOTA success rate of 31.9%.

CER learns experience in an online style on WEBARENA (Zhou et al., 2024b) and VISUALWE-BARENA (Koh et al., 2024a) for lack of training set and fair comparison. We also did an extensive analysis to extend CER to offline and offline and online hybrid settings (§5.1). We found CER works well with the offline data provided. It can even improve further with a limited number of human-annotated trajectories to warm up before online learning.

We further investigate the improvements of CER with various metrics, such as cross-template success rate, stability (preservation of old knowledge), and plasticity (acquisition of new knowledge) (Grossberg, 1982; Rolnick et al., 2019) (§5.2, §5.3), demonstrating its generalizability and effectiveness as a continual learning system. Also, we show that through the combination with a sampling-

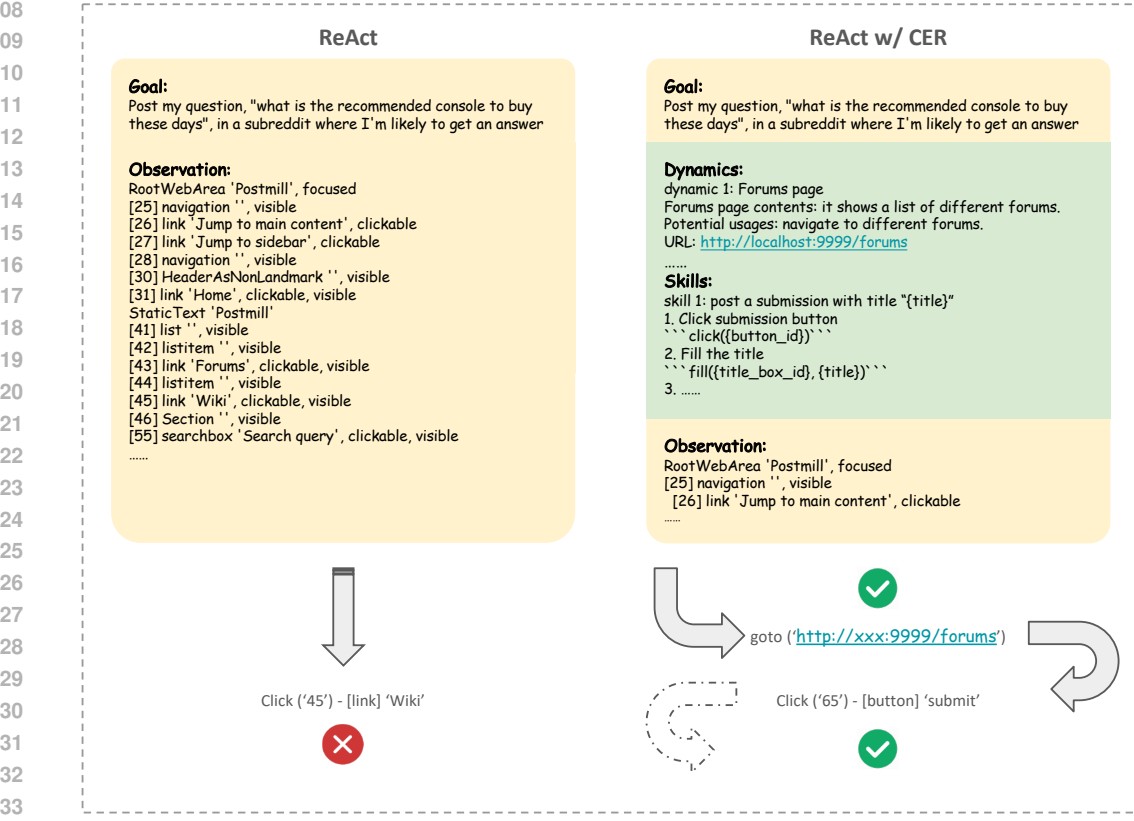

Figure 2: Compare ReAct baseline with ReAct + CER. The experiences, including dynamics and skills, are obtained through multiple modules as in Fig.1. They are "replayed" in the context window of the model, helping the agent to make correct decisions. For simplicity, the thinking process is neglected in the figure.

based method, CER pushes the boundary forward again, showing its compatibility with other methods (§5.4).

In summary, our contributions are as follows:

- We propose a simple but effective continual learning framework for language agents: CER. CER distills fine-grained skills and environment dynamics from both successful and failed trajectories. Importantly, it works for offline, online, and hybrid settings.

- CER shows state-of-the-art performance on multimodal web navigation tasks. It also shows excellent stability and plasticity (Grossberg, 1982; Rolnick et al., 2019), as well as good synergy with other off-the-shelf methods.

- We do a comprehensive analysis on CER to prove its validity and understand the improvements better.

## 2 RELATED WORK

**LLM Agents**  The increasing capabilities of LLMs have enabled new applications where agents built with LLMs can take action and interact with external environments. To enable action-taking, methods like ReAct (Yao et al., 2023b) prompt LLMs to interleave actions and reasoning in the output. Apart from action-taking, planning and search is also an important component for agents. Methods like Reflexion (Shinn et al., 2023), Self-Refine (Madaan et al., 2023), Tree-of-Thought (Yao et al., 2023a), and Tree Search (Koh et al., 2024b) (Zhou et al., 2024a) enable LLMs to revise their reasoning and perform deliberate search among their action space.

**Web Agent Environments** LLM agents are increasingly being employed to perform various digital tasks on behalf of humans, with interacting with websites being a common application area supported by numerous benchmarks. For instance, WebShop (Yao et al., 2022) tasks agents with identifying products that meet specific user requirements by interacting with e-commerce platforms. Extensions such as WebArena (Zhou et al., 2024b) and Mind2Web (Deng et al., 2023) have broadened the scope of tasks to include a wider variety of websites and more realistic applications, encompassing activities like trip booking, information retrieval, website navigation, and social media management. VisualWebArena (Koh et al., 2024a) designs challenging multimodal web navigation tasks that require agents to leverage visual grounding and understand image inputs. Among these benchmarks, WebArena and VisualWebArena provide the most realistic, controllable, and interactable environments, which makes the tasks more challenging and the results reproducible. The interactive characteristics are also beneficial for our continual learning paradigm.

**Learning from Memory or Past Experiences** Some previous works have investigated the storage of memories of past agent trajectories. Generative agents (Park et al., 2023) use similar strategies to investigate human behaviors with such a human-like strategy. Voyager (Wang et al., 2024a) enables the agent to learn diverse skills in Minecraft.

Similarly, frameworks such as ExpeL (Zhao et al., 2023) and Synapse (Zheng et al., 2023) leverage stored past task trajectories as memory, which are dynamically retrieved to support task execution. However, they either test on relatively simple web environments (Yao et al., 2022) or use raw and long observation-action pairs as exemplars directly, which limits their applicability to more complex environments. In a concurrent work Agent Workflow Memory (Wang et al., 2024b), they also propose the idea of summarizing workflows from past trajectories and augmenting the agent with the workflows. However, only successful trajectories are considered. Additionally, they do not have a retrieval module and only have a summarization module that summarizes all past trajectories each time and updates the whole workflow memory in a rewriting style, which hinders the accumulation of experiences and limits the applicability of their method to a more continual paradigm. In our work, we construct a well-designed, efficient, and scalable continual learning framework for autonomous language agents and test it in two challenging and realistic web environments. The experiences contain both environment dynamics and decision-making patterns. We also investigate it both qualitatively and quantitatively and demonstrate its advantage in terms of applicability in different learning paradigms and compatibility with other agent methods.

## 3 CER: Contextual Experience Replay

Consider a general setup for a language agent $A$, powered by a language model $M$ with a context window $C$, to solve a sequential decision-making task in an environment. CER include four separate modules: distillation module $D$, retrieval module $R$, dynamic experience buffer $\epsilon$ and the base decision-making agent itself $A$ as shows in Fig.1. CER can start working given a arbitrary set of trajectories $\mathbb{T} = \{\tau_1, \tau_2, \ldots, \tau_n\}$. All modules here are implemented by prompting a visual language model (VLM), i.e. GPT-4o in our implementation. Details of prompts for each module can be found in A.1.

### 3.1 Distill experiences from trajectories

Given a trajectory set $\mathbb{T}$, the distillation module will distill experiences $\mathbb{E} = \{E_1, E_2, \ldots, E_n\}$ from them one by one where $E_i = (D_i, S_i)$. $D_i$ stands for environment dynamics, or dynamics in short, and $S_i$ represents useful decision-making patterns, or skills in short. The dynamics provide useful state information to help the agent make state-aware decisions or directly navigate to the state through its URL. These skills provide common decision-making patterns, inspiring agents to take better action. We use two separate modules for the distillation of dynamics and skills due to their different characteristics. The output format is similar to ReAct (Yao et al., 2023b), asking the model to issue a think action before outputting each distillation. The dynamics distillation module will distill a list of summaries of different web pages, their corresponding URL, and inferred possible usages. The skill distillation module is instructed to summarize a list of useful skills. Each of them includes a brief overall summary (e.g. Navigate to forum {forum name}) and the corresponding detailed step-by-step guidelines. Specifically, the guideline contains both natural language summaries and

concrete action examples for each step, as the example in Fig.2. While the natural language summaries provide flexible and general high-level instruction, the example helps the agent to understand the step and also format its output better. The model is required to output the final distillation in an abstract and general way, i.e. navigate to forum {forum name} instead of navigate to forum "books", to ensure that the experiences can be broadly applied. The model is also provided with existing experiences in the buffer to avoid repetitive distillation, allowing the continual accumulation of the experiences across time.

## 3.2 RETRIEVE EXPERIENCES FROM BUFFER

After the distillation period, the buffer $\epsilon$ now includes a set of useful experiences $\mathbb{E} = \{E_1, E_2, \ldots, E_n\}$. Similar to the distillation module, we designed two separate modules to retrieve dynamics and skills correspondingly also using ReAct style output format (Yao et al., 2023b), additionally prompting with general instructions, the current task goal, the website descriptions, and all dynamics or skills available in the buffer. Then, the module will retrieve the top-$k$ useful and informative experiences from the buffer by their ids and pass this to the language agent. This module makes it possible for the distillation module to continuously merge new experiences and help the agent filter out useful experiences for the current task.

## 3.3 DECISION-MAKING WITH CONTEXTUAL EXPERIENCE REPLAY

To best utilize the in-context learning capability of language models, we transform the selected $k$ experiences $\mathbb{E} = \{E_1, E_2, \ldots, E_k\}$ into natural language experience descriptions $E_{NL} = f(\mathbb{E})$ through a programmatic mapping f and integrate them into the model's context $C$, resulting in a new augmented context $C' = g(C, E_{NL})$. Therefore, the decision-making policy underneath will be influenced by the additional experiences, and the agent $A$ can issue better actions with reference to the experiences. The context comparison between the baseline agent and CER is shown in Fig.2.

## 3.4 COMBINATION OF OFFLINE AND ONLINE LEARNING

The source of the trajectories to learn from is important for CER. Depending on the source of trajectory data, CER can be divided into offline, online, and hybrid versions. Online data is collected from past task-solving trajectories in the environment during inference time. Specifically, in the online setting, there are no trajectories provided at the very beginning, but as the procedure goes on, there will be self-generated trajectories from past tasks. CER will run the distillation module after each task and run the retrieval and replay module in the next task. Different from an online setting, offline learning means there is a training set of trajectories at the beginning for CER to learn from but no further learning during inference. Additionally, these two settings can be combined to serve as a whole system, i.e., learn from a fixed training set first and then self-evolve in the environment with self-generated data. We investigate the effectiveness of these two settings in the section 5.1.

## 4 EXPERIMENTS

We evaluate CER on the full set of WEBARENA (Zhou et al., 2024b) (WA) and VISUALWEBARENA (Koh et al., 2024a) (VWA) in online setting for fairness because training set is not provided in these benchmarks. Also, it would be interesting to see how CER performs without additional external data in a close-loop continual learning paradigm. We did further studies about offline and offline + online settings in section 5.1. The reason we chose these two is that they provide interactive, realistic, and reproducible web environments that are better for applying continual learning and still close to real-world scenarios. WEBARENA have 812 tasks across five different websites corresponding to different domains: shopping, shopping administration, online forum, map, and project collaboration (Gitlab). VISUALWEBARENA retains the shopping and forum website, adds another classifieds website, and designs 910 tasks on top of them. Although they share two websites, the focus of their tasks is different. Most of the tasks in WebArena only have text descriptions of task goals, while a large portion of tasks in VisualWebArena involve visual input as part of task goals and require an understanding of the visual information of the current website. This also leads to their large variations of task types.

Table 1: Success rates (SR) of published open-source methods and CER up to the completion of this work on WEBARENA, **Bold** represents the best result on the website while underline means the second best results. CER uses text observations (accessibility tree) only and $CER_v$ takes both text and visual observations. The results originate from the corresponding papers except BrowserGym which we reproduce the GPT-4o version by ourselves. *: SteP (Sodhi et al., 2024) uses human-designed detailed policies for each website, so it is not comparable with other autonomous methods without human involvement and we set it apart just for references.

| Method | Shopping | CMS | Forum | Gitlab | Map | **Average** |
|---|---|---|---|---|---|---|
| SteP* (Sodhi et al., 2024) | 37.0 | 24.0 | 59.0 | 32.0 | 30.0 | 33.0 |
| WebArena (Zhou et al., 2024b) | 24.0 | 11.0 | 7.9 | 10.2 | 21.1 | 15.0 |
| AutoEval (Pan et al., 2024) | 25.5 | 18.1 | 25.4 | 28.6 | **31.9** | 20.2 |
| BrowserGym (Drouin et al., 2024) | 26.6 | 28 | 22.8 | 21.4 | 18.4 | 24.3 |
| CER (ours) | **33.5** | 33.7 | 30.7 | 29.1 | 30.4 | 31.4 |
| $CER_v$ (ours) | 29.2 | **36.3** | **37.7** | **34.2** | 28.6 | **33.2** |

## 4.1 IMPLEMENTATION DETAILS

### 4.1.1 WEBARENA

For WebArena, all tasks in it provide text-only task instruction, so we implement two versions of CER; CER takes text-only observation, and $CER_v$ takes both text and visual observation of the environment. The text observation is an accessibility tree representation of the current webpage, and the visual observation is a screenshot of the current page. For both versions, we use GPT-4o-2024-0513 as the backbone language model with a temperature of 0.1. We use BrowserGym (Drouin et al., 2024) as the environment, which provides both text and visual observation for the agent and adds additional information for clickable and visible elements in the accessibility tree of the webpage. To fairly highlight the improvement of CER, we run GPT-4o w/ BrowserGym (Drouin et al., 2024) by ourselves as the baseline for comparison. CER is compatible with most off-the-shelf language model agents since it only needs the past trajectories. Here, we test it with a simple method by prompting GPT-4o directly and using ReAct (Yao et al., 2023b) as the output format as in BrowserGym (Drouin et al., 2024) and WebArena (Zhou et al., 2024b). We also combine it with another performant method and observe significant improvements(§5.4). We set the retrieval parameter to $k_d = 5$ and $k_s = 5$, denoting the maximum number of dynamics/skills to retrieve and replay.

### 4.1.2 VISUALWEBARENA

For VISUALWEBARENA, similar to WEBARENA, we still use BrowserGym (Drouin et al., 2024) as our environment. Since BrowserGym does not support visual evaluation, we implemented the environment by ourselves and built CER on top of that. We also run BrowserGym results as the baseline for comparison. Using the same setting as (Koh et al., 2024a), we apply Set-of-Marks (SoM) (Yang et al., 2023) to the original screenshot of the webpage. This method marks each interactable element of the webpage with a highlighted bounding box and the corresponding unique element ID on the corner of the box to enable grounding. Besides the screenshot, the agent is also provided with text observation of the environment for better grounding, where the ID of each element is consistent with the one in the SoM-processed screenshot.

## 4.2 RESULTS

Our results on these two benchmarks are summarized in Table 1 and Table 2. On WEBARENA and VISUALWEBARENA, while orthogonal to the other methods, CER achieves state-of-the-art performance and improves the baseline agent, GPT-4o w/ BrowserGym (Drouin et al., 2024), relatively by 36.6% and 21.8% respectively. It should be noted that SteP (Sodhi et al., 2024) uses human-designed

Table 2: Success rates (SR) of published open-source methods and CER on VISUALWEBARENA, **Bold** represents the best result in the domain. Results are from the corresponding papers except BrowserGym. We implement the agent with BrowserGym by ourselves.

| Method | Classifieds | Shopping | Forum | **Average** |
|---|---|---|---|---|
| VisualWebArena (Koh et al., 2024a) | 18.4 | 20.0 | 17.1 | 18.9 |
| BrowserGym (Drouin et al., 2024) | 26.2 | 28.2 | 21.1 | 26.2 |
| Tree Search (Koh et al., 2024b) | 26.5 | 29.0 | 20.5 | 26.4 |
| CER (ours) | **27.0** | **38.1** | **24.4** | **31.9** |

Table 3: Success rates (SR) of different settings with different offline data source on the Forum tasks split of the WebArena.

| | Offline data source | SR |
|---|---|---|
| Baseline | - | 22.8 |
| Offline | human annotations | 33.3 |
| | self-guided explorations | 31.6 |
| Online | - | 37.7 |
| Offline + Online | human annotations | **41.2** |
| | self-guided explorations | 35.1 |

policies, i.e., step-by-step instructions for each website split, and can need much extra human effort when encountering new cases or on new websites. So we do not consider it when comparing CER with other methods. On VisualWebArena, CER achieves SOTA performance and outperforms the tree search method (Koh et al., 2024b), which is also built on GPT-4o, with much lower token costs. The result of the tree search is obtained through a search algorithm that uses 20 times sampling at each step, and a maximum of 5 steps, with extra costs of GPT-4o used as a value function. In our implementation, we use a maximum of only 30 steps for each task, similar to the setting in (Zhou et al., 2024b) and (Koh et al., 2024a), thus using at least 3 times fewer tokens.

## 5 ANALYSIS

In this section, we conduct experiments on offline and offline + online hybrid settings of CER to show its potential with a few offline trajectories. Furthermore, we conduct extensive analysis to investigate and better understand CER's improvements through cross-template success rates and two interesting metrics for continual learning systems: stability and plasticity. Finally, we validate its compatibility and synergy with other performant methods, proving its wide applicability.

### 5.1 ONLINE AND OFFLINE LEARNING

As mentioned in section 3.4, CER can also be run in offline or offline+online hybrid settings. We conduct offline and offline + online experiments on the Forum tasks split of WebArena with two sources of offline training trajectories: from human demonstrations or from self-guided explorations.

For the human annotation data, We designed five tasks, which were validated as not appearing in the test set, and denoted corresponding oracle trajectories for Forum websites. For the random exploration data, we prompt a language model to propose diverse actions at each step and collect the final exploration trajectories. The details of prompts and tasks can be found in A.3 and A.4. The overall results are shown in Table 3. Both training sources improve the performance over the baseline in the offline setting. Notably, with five human annotations as the training set, the

performance of offline + online learning surpasses the original online learning. To understand how offline and online learning synergize with each other, we take task 31 as an example: the agent is asked to get the count of comments that have received more downvotes than upvotes for the user who made the latest post on the photoshopbattles forum. In online settings, the agent does not know how to sort the posts on the Forum website either because this task is at the beginning, and it has not learned many experiences yet. However, from the training set, CER distilled the page summary of the forums page where all forums are displayed and also the skill of sorting the list of posts. Aware of the existence of the forums page, the agent knows to click the "forums" button to navigate to the list of forums. After that, it is inspired by the skill and sorts the posts correctly. It finishes the task successfully with dynamics and skills distilled from the training set.

Although in the offline setting, both training sources help the agent outperform the baseline, offline + online settings with self-guided explorations perform even worse than the online-only setting. We analyzed the results and found that the trajectories collected through such explorations are highly unstructured and noisy. The exploration agent can jump from one action to another unrelated one. So, the distillation module can hardly distill useful patterns from it. So, the distilled experiences may even mislead the agent sometimes.

In real-world scenarios, high-quality human-annotated training data is hard to collect, so online learning is still important and meaningful in most cases. It would be interesting to explore the potential of CER with more high-quality human-labelled trajectories. The negative impact of the training set derived from explorations also indicates that goal-oriented trajectory matters for CER because it has more structured, continuous, and relatively meaningful action sequences rather than unordered small pieces.

## 5.2 INVESTIGATING IMPROVEMENTS OF CER

In this section, we try to understand where the improvements of CER come from and get some intuitions about how CER works.

Intuitively, the state space and action space for the current step are extremely large. However, for human users, the states that we often navigate to and the actions that we usually take are only a small subset of the whole space. Experiences distilled from some goal-oriented trajectories tend to contain some informative and effective states and actions that are often navigated to or used. With the highlighted promising states, actions, and decision-making patterns, the agent can issue a correct action much more easily. Of course, some of the experiences can be noisy and misleading. We show in section 5.5 that CER is still robust to the correctness of the trajectories. This intuition also aligns with the Recognition Primed Decision making Model proposed by Klein (1998), where humans tend to recognize promising actions when encountering complex environments.

We also conduct quantitative comparisons between CER and baseline method to investigate the improvements. The tasks in WebArena are designed based on templates, and at most, five tasks share the same template. For example, What is the top-1 best-selling brand in Quarter 1 2022 is built based on the template: What is the top-n best-selling brand in period. Although many tasks do not share exactly the same problem-solving pattern, if the agent just memorizes the pattern of the whole task, it will be able to solve some other tasks in the same template more easily, thus improving the overall performance. So, we use the cross-template average success rate, calculated by the number of templates solved (at least one task is solved) divided by the total number of templates. We run experiments on Forum tasks of WebArena. The results are shown in Table 4. CER shows a significant improvement in cross-template success rates. This result validates that the improvement of CER does not come from memorizing the whole trajectory of a task. Instead, it distills more fine-grained experiences, which allows for the generalization of different types of tasks. Also, we analyze the improvements from the perspective of stability and plasticity, which measures the ability to retain original ability and learn new things. More details are in section 5.3.

## 5.3 STABILITY AND PLASTICITY

A well-designed continual learning system should demonstrate both stability (preservation of old knowledge) and plasticity (acquisition of new knowledge) (Grossberg, 1982; Rolnick et al., 2019). Since knowledge is hard to measure in our case, we measure the acquisition of new knowledge

Table 4: Cross-template success rates (ct-SR), stability and plasticity of CER and baseline on the Forum split of WebArena

| Method | ct-SR | Stability (%) | Plasticity (%) |
|---|---|---|---|
| Baseline | 44.7 | 100 | 100 |
| CER | **60.0** | 93 | 141 |

through problem-solving ability, i.e., success rates, in a specific environment. Therefore, we similarly measure the stability and plasticity of CER in cross-template success rate (ct-SR) since the success in new types of task demonstrates new problem-solving ability. Specifically, stability is measured through the percentage of tasks from the baseline that CER is able to solve, which reflects how well CER maintains the original capability of the baseline. Plasticity is measured by the improvement of CER on new cases, measuring how many additional tasks CER can solve compared to the baseline. Since CER can be understood as a continual learning system built on the baseline method agent. We set the stability and plasticity of the baseline to 100% and used the ct-SR to calculate the stability and plasticity of CER. The results are also shown in Table 4. With most of the original abilities retained, CER demonstrates 41% new problem types solved, proving the validity of CER as a continual learning framework. This also indicates the potential of the compatibility with other performant methods, which we discuss in detail in section 5.4.

## 5.4 Synergy with performant methods

We also analyze the compatibility of CER with other performant methods. Tree search (Koh et al., 2024b) is a computing-intensive method with more explorations and backtracking to search for better action to take. However, due to the high costs and the long time it takes, we chose another comparable method of it: trajectory sampling and reranking. We sampled 3 times for each task with max steps of 20 and prompted a language model with the trajectories to give a score and select the trajectory with the highest score as the final one. The procedure of applying CER to such method is similar to what we do with baseline agent. We conduct experiments in an online setting on the Forums split of WebArena. The results are in Table 5. CER with sampling method improves CER w/ ReAct performance by a relative success rate increase of 39.5%. This is because, firstly, such performant methods generally have better precision from start to end, so the experiences distilled from them

Table 5: Comparison of success rates (SR) of CER and CER w/ trajectory sampling and reranking on the Forum split of WebArena

| Method | SR |
|---|---|
| CER | 37.7 |
| Sampling | 43.1 |
| CER w/ sampling | **52.6** |

are of higher quality. Additionally, high-quality experiences make performant methods more robust through the learned experiences and provide environment-specific knowledge to help better decision-making. Also, for exploration methods like tree search, CER should enable it to learn from past experiences and avoid searching from scratch each time, lowering its costs significantly.

## 5.5 Access to ground truth rewards

Currently, CER distills experiences from both successful and failed cases. To investigate whether the distillation from failed cases is a bottleneck of CER, we run CER for only successful trajectories evaluated by ground truth evaluators. We conduct the experiments on the full set of WebArena. The results are summarized in Table 6. The results show that CER performs better with the access to ground truth reward. The possible reason could be that the successful trajectories have higher quality and are more informative, while the failed trajectories provide some misleading action sequences that will negatively influence the agent's decision-making.

Nevertheless, the acceptable gap and significant improvements over the baseline agent show the robustness of CER given noisy trajectories. This gives credit to the implicit reasoning ability and the flexible natural language representation of experiences. Although provided with a few noisy

Table 6: Success rates (SR) of CER and CER$_{success}$ on five website splits of the WebArena. CER is the main method we used before, which learns from both successful and failed trajectories. CER$_{success}$ uses ground truth evaluators from the environment to filter out and learn from successful experiences only. Both method takes text observation for comparison

| Method | Shopping | CMS | Forum | Gitlab | Map | **Average** |
|---|---|---|---|---|---|---|
| CER | **33.5** | 33.7 | 29.1 | 30.7 | **29.1** | 31.4 |
| CER$_{success}$ | 32.8 | **40.6** | **31.6** | **33.7** | 25.0 | **33.5** |

Table 7: Success rates (SR) of CER on the Forum split of WebArena with different ablation settings to the experiences.

| Method | SR |
|---|---|
| CER | **37.7** |
| CER- skills | 33.3 |
| CER- dynamics | 35.1 |

experiences, the agent can still filter out useful trajectories and issue reasonable action most of the time.

### 5.6 DIVISION OF DYNAMICS AND SKILLS

We conduct an ablation experiment to investigate the necessity of the division of dynamics and skills. The experiment is run on the Forums split of WebArena. The results in Table 7 show that both dynamics and skills are important for CER. Environment dynamics make the agent aware of the content of many pages and also provide the URL to navigate to. Skills inspire the agent and also provide promising actions to be taken in the current step. They provide heuristics in terms of states and actions correspondingly and synergize with each other.

### 5.7 LIMITATIONS AND FUTURE WORK

Despite the substantial progress achieved with CER, there are several limitations that will influence its applicability and could be addressed in future work. First, section 5.1 shows that although CER performs even better in offline + online settings, it requires the trajectories to be goal-oriented to distill high-quality experiences. The performance is limited if trajectories from random explorations are provided. The more fine-grained utilization of low-quality trajectories could be explored in the future. Secondly, the environment dynamics help much in web environments, as shown in section 5.6, partially because the agent can directly navigate to a specific page with its URL. It would be interesting to investigate how we can utilize the environment dynamics in other agent tasks, such as real-world navigation (Shridhar et al., 2021) in future work.

## 6 CONCLUSIONS

In this paper, we proposed a training-free framework for efficient and effective continual learning of language agents in complex web environments. Our framework enables language agents to learn from past experiences and replay during inference time for better decision-making. We also conduct extensive analysis to investigate the potential of CER under various different settings, including offline or offline + online paradigm. Furthermore, we validate the effectiveness of it as a continual learning system through stability and plasticity. We believe that learning from past experiences is crucial for building a helpful computer agent that can adapt to different environments and evolve autonomously.

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

# A   APPENDIX

## A.1   CER PROMPTS

Here we provide detailed prompts for each module in CER. Figure 3 and 4 show the system prompts for distillation modules, while Figure 5 and 6 are system prompts for retrieval modules.

## A.2   EXPERIENCE EXAMPLES

We provide experiences examples for WEBARENA and VISUALWEBARENA here. These natural language experiences are programmatically transformed from the structured experiences stored in the experience buffer and will be added to the system prompt of the model when solving tasks. The examples are shown in Figure 7 and 8.

## A.3   SELF-GUIDED EXPLORATION PROMPT

As mentioned in section 5.1, we prompt a language model, i.e., GPT-4o, here to explore diverse actions in the environment and collect the corresponding trajectories. We limit the max steps to 30 and sample 10 trajectories. The instruction part of the prompt is shown in Figure 9.

## A.4   HUMAN ANNOTATED TASKS

As stated in section 5.1, we designed 5 tasks on the Forum website and asked a human annotator to annotate the oracle trajectory for each task. The full list of task instructions is as follows:

- *Find the title of the most commented post in forum 'history'.*
- *Find the title of the most controversial post of all time in forum 'Paterson'*
- *Go to the user's profile who made the comment "What are you doing in a deep learning sub?"*
- *Upvote the comment of the current user which replies to 'Maybe there's just something wrong with me.' by HedleyLamarrrr*
- *Reply to the first comment of the post titled 'It's the only logical explanation' with 'lol'*

You will be given the state-action trajectory of a user interacting with a webpage and the overall goal of the trajectory. You need to summarize the useful pages and pair it up with the corresponding URLs.
Output format:
<URL>
the URL of page 1
</URL>
<think>
think step by step like in the examples and summarize the page
</think>
<page-summary>
the brief summary of page 1, following the format:
Name: name page
Description: descriptions
Usages: usages
</page-summary>
<URL>
the URL of page 2
</URL>
<think>
think step by step like in the examples and summarize the page
</think>
<page-summary>
the brief summary of page 2, following the format:
Name: name page
Description: descriptions
Usages: usages
</page-summary>
...

# Examples
## Example 1
Overall goal of the trajectory: Go to r/books forum.
Current website: Reddit
Existing summarized pages:
Page 1: Profile page
Description: it shows the user's profile information.
Usages: view or modify user's profile information.
URL: https://www.example.com/profile
Human user trajectory: [neglected here]

## Output:
<URL>
https://www.example.com/forums
</URL>
<think>
From the content of the page, it shows a list of forums, I can summarize it to Forum page. The website is Reddit so this page can be used to navigate to different forums.
</think>
<page-summary>
Name: Forums page; Description: it shows a list of different forums.; Possible usages: navigate to different forums.
</page-summary>

IMPORTANT NOTES you should absolutely follow:
1. DO NOT include any other words except url, think and page summary as the format stated above.
2. Follow the example to think and summarize the page.
3. You should only summarize once for each unique URL.
4. Check existing pages before generating, do not summarize pages that have already been summarized, instead, use "Summarized before" in the steps.
5. Focus on the main content of the page and may ignore the modifications made by the user when generating the summary.

Figure 3: System message for dynamics distillation module in CER

810
811
812
813
814
815
816
817
818
819
820
821
822
823
824
825
826
827
828
829
830
831
832
833
834
835
836
837
838
839
840
841
842
843
844
845
846
847
848
849
850
851
852
853
854

You will be given the state-action trajectory of a user interacting with a webpage and the overall goal of the trajectory.
You need to summarize skills from the trajectory.
Skills are a subset of actions that the user takes to achieve a sub-goal.
You should break the overall goal into sub-goals and summarize each sub-goal as a skill.
Represent the non-fixed elements (input text, button strings) and non-fixed words (e.g. a specific forum name / user name; an option) with descriptive variable names as shown in the example.
Output format:
<think>
think step by step
</think>
<skill>
skill1 name here.
</skill>
<steps>
The steps of the skill1 here.
</steps>
<think>
think step by step
</think>
<skill>
skill2 name here.
</skill>
<steps>
The steps of the skill2 here.
</steps>
...
# Examples
## Example 1
Overall goal: I want to get the cheapest product in the Cabinets, Racks & Shelves category
Current website: current website
Existing skills:
Skill 1: Sort products by sort criterion
1. To sort the products by sort criterion, I need to click on the "Sort by" dropdown menu.
"'click(sort by id)'"
2. To sort the products by sort criterion, I need to select the sort criterion option from the "Sort by" dropdown menu.
"'click(sort criterion id)'"
Human user trajectory: [neglected here for length]
##Output: [neglected here for length]
IMPORTANT NOTES you should absolutely follow:
1. DO NOT include any other words except skills and steps as the format stated above.
2. Check existing skills before generating; do not summarize skills that have already been summarized; instead, use "Summarized before" in the steps.
3. You should break the overall goal into sub-goals and summarize each sub-goal as a skill.

855
856
857
858
859
860
861
862
863

Figure 4: System message for skills distillation module in CER

You will be given a goal of a task to be executed on a website and a list of urls and the corresponding page summary to choose from.
You need to select the pages that most possibly need to be visited to achieve the goal.
You should break the task down into a few steps so that you can select the pages that can help most in each step.
IMPORTANT: You should select not more than max skills pages!
Output format:
<think>
think step by step. Break the task down into a few steps and then select pages
</think>
<selected-pages>
id: the id number (the number at the beginning) of page 1; name: page 1 name
id: the id number (the number at the beginning) of page 2; name: page 2 name
...
</selected-pages>

# Examples
## Example 1
Task goal: Upvote the hottest post in r/books
Current website: website descriptions
Shortcuts to choose from:
id: 1; name: Forums page; description: It shows a list of different forums; possible usages: navigate to different forums; url: https://www.example.com/forums
id: 2; name: Profile page; description: It shows the information of current user; possible usages: Check or modify the information of the current user; url: https://www.example.com/profile
id: 3; name: Submission page; description: It provides a few text boxes to fill in to submit a new post; possible usages: Submit new posts; url: https://www.example.com/submission
id: 4: name: Subscribed forums page; description: It provides a list of subscribed forums; possible usages: check or navigate to subscribed forums; url: https://www.example.com/subscribed
## Output 1:
<think>
The goal is to upvote the hottest post in r/books. The user needs to navigate to the r/books page first or go to forums to find the r/books page. Then the user needs to find the hottest post in the r/books page. So the useful pages from the shortcuts are Forums page
</think>
<selected-pages>
id: 1; name: Forums page
</selected-pages>

Figure 5: System message for dynamics retrieval module in CER

You will be given a goal of a task to be executed on a website and a list of skills to choose from.
You need to select the skills that can help most in achieving the goal.

You should break the task down into a few steps so that you can select the skills that can help most in each step.
IMPORTANT: You should select not more than max skills skills!
Output format:
<think>
think step by step. Break the task down into a few steps and then select skills
</think>
<selected-skills>
id: the id number (the number at the beginning) of skill 1; name: skill 1 name
id: the id number (the number at the beginning) of skill 2; name: skill 2 name
...
</selected-skills>
# Examples
## Example 1

Task goal: Upvote the hottest post in r/books
Current website: website descriptions
Skills to choose from:
Skill 1: Navigate to forums
1. Click on the "Forums" menu item.
"'click(forums id)"'
2. Click on the specific forum name.
"'click(forum name id)"'
Skill 2: Submit a new post
1. Type the post title in the title text box.
"'type(title text box id, "Post Title")"'
2. Type the post content in the content text box.
"'type(content text box id, "Post Content")"'
3. Click on the "Submit" button.'
"click(submit button id)"'
Skill 3: Sort posts by sort criterion
1. Click on the "Sort by" dropdown menu.
"'click(sort by dropdown id)"'
2. Select the sort criterion option from the "Sort by" dropdown menu.
"'click(sort criterion id)"'

Output:
<think>
The goal is to upvote the hottest post in r/books. The user needs to navigate to the r/books page first or go to forums to find the r/books page. Then the user needs to find the hottest post in the r/books page. So the useful skills from the shortcuts are Navigate to forums, Sort posts by hotness
</think>
<selected-skills>
id: 1; name: Navigate to forums
id: 3; name: Sort posts by hotness
</selected-skills>

Notes:
1. Some skills might not be consistent with the current task but it is still useful to refer to, e.g. write a post to express happiness is useful in a task to write a post to express sadness.

Figure 6: System message for skills retrieval module in CER

---

# Environment dynamics (common pages to navigate to from previous experience):
## Dynamics 1: Forums page
### Forums page contents: it shows a list of different forums.
### Potential usages: navigate to different forums.
### URL: http://localhost:9999/
...
# Skills (common workflows summarized from previous experience):
## Skill 1: Edit profile biography
### Steps:
1. Navigate to the Edit Biography page.
"'goto('http://localhost:9999/user/username/edit biography')'"
2. Fill in the biography text area with the new biography content.
"'fill(biography text area id, 'new biography content')'"
3. Click on the save button to update the biography.
"'click(save button id)'"
...

Figure 7: Experience snippet example on WEBARENA

---

# Environment dynamics (common pages to navigate to from previous experience):
## Dynamics 1: Video Games category page
### Video Games category page contents: This page lists products that are within the "Video Games" category, including accessories, consoles, and other related products
### Potential usages: Browse and purchase video game-related products.
### URL:http://localhost:7770/video-games.html
...
# Skills (common workflows summarized from previous experience):
## Skill 1: Sort search results by {sort criterion}
### Steps:
### 1. Open the sort dropdown menu by clicking on the sort dropdown button.
"'click({sort dropdown button id})'"
2. Select the "{sort criterion}" option from the sort dropdown menu.
"'select$_o$ption({sortdropdownbuttonid}, {sortcriterion})'"
...

Figure 8: Experience snippet example on VISUALWEBARENA

---

# Instructions: Your objective is to discover diverse and interesting tasks (that a human might give to an agent) by interacting with the webpage through these actions. You've executed the following actions, and observed the following webpage states [observations and other information are neglected here]

Figure 9: The instruction part of prompt for random explore agent

