# OpenReview forum: "Contextual Experience Replay for Continual Learning of Language Agents"
_ICLR.cc/2025/Conference — Submitted to ICLR 2025_

### Official Review · Reviewer_FBmG · 2024-10-19

**Soundness:** 2
**Presentation:** 2
**Contribution:** 2
**Rating:** 3
**Confidence:** 5

**Summary:**

This paper introduces the concept of Contextual Experience Replay (CER), allowing agents to leverage limited experiences while performing web navigation tasks. These experiences can be categorized as offline, which are pre-defined external experiences, online, where the agents reuse their own previous experiences, or a hybrid of both. The authors evaluated CER on two representative web navigation tasks, demonstrating its effectiveness.

**Strengths:**

The idea proposed in this paper has practical significance. Considering the high post-training costs, a training-free self-improvement method is undoubtedly more attractive, yet also more challenging.

The authors have made some commendable choices in their experimental setup. For instance, they thoroughly considered offline, online, and hybrid scenarios, resulting in a well-rounded narrative. Furthermore, the ablation study was quite comprehensive, examining aspects like template stability, plasticity, etc., which provided a better characterization of CER.

The paper's layout, illustrations, and tables are aesthetically pleasing, leaving a good impression on the readers.

**Weaknesses:**

Nonetheless, I must offer my apologies as I have several concerns regarding this paper. Allow me to expound on each of these concerns.

First, from a writing standpoint, the paper gives me a somewhat informal impression. (Although the layout is visually appealing, the writing itself, in my opinion, is not of a satisfactory standard.) For instance, in the introduction, the authors emphasize the advantages of web navigation tasks, such as being more controllable than OS tasks and more complex than mobile environments. However, this does not justify why the authors only tested on two related tasks from WebArena in this work on CER. Moreover, considering that Visual WebArena involves multimodal capabilities (which is commendable for incorporating such capabilities), it effectively means that one pure language task and one multimodal task were tested, making it difficult to be convinced of the generalization of the CER method. For example, at the very least, attempting a classic benchmark like WebShop would make this work more persuasive.

In terms of writing, I feel that the paper uses too many terms without sufficient explanation, which gives the impression of inundating readers with jargon. For example, the authors define their work as continual learning and experience replay. If these terms are well-established in the field, the theoretical nature of these terms warrants the establishment of a theoretical framework to prove the effectiveness of the CER method. Conversely, if these terms are coined by the authors, more rigorous discussion is needed rather than presenting the terms without further elaboration, which leaves a sense of shallowness.

Incidentally, I also mentioned that this paper's idea merits a theoretical framework for analysis. I recommend referring to the following papers:

https://arxiv.org/pdf/2407.18219

https://openreview.net/forum?id=9W6Z9IeLzc

Next, in terms of experiments, I have already pointed out the deficiencies of the benchmark. Nevertheless, it appears that the authors primarily tested the effectiveness of GPT-4o, a single model, and conducted most of the experiments based on this closed-source model? This raises several concerns, such as the possibility of real-time updates to the closed-source model, making it difficult to reproduce the results in future work. Additionally, can using only one current state-of-the-art model demonstrate that CER is also effective for weaker open-source models? Therefore, I strongly recommend that the authors conduct more experiments using a wider variety of models. If the authors are unable to do so due to the high cost of experiments, I fully understand. However, I did not see such a statement in the paper. (I believe you could have used this reason to justify why only web navigation was selected since the academic community understands the scarcity of computing resources on campus. However, given your previous claim that web navigation is more controllable and complex, it is hard to avoid the impression that you are not lacking resources but are unwilling to put in the extra effort to illustrate your method more comprehensively.)

Lastly, in terms of method, I do not entirely agree that CER is the first framework to leverage experiences. At least, you can refer to this 2023 paper, which proposed the idea of leveraging experiences earlier:

https://github.com/PanasonicConnect/rap

Please refrain from overstating your work, or at least add qualifiers to narrow the scope of your research.

There are also some debatable aspects, such as your reliance on relative improvement in your experimental results. In my view, this is not an ideal metric because, as you mentioned, these two web navigation benchmarks only have a success rate of less than 20%. Even if you achieved more than a 30% relative improvement, how significant is this improvement in absolute terms?

In summary, I am pleased that you recognize the importance of experience for workloads like agents, which is closely related to in-context learning in NLP. I believe you could also attempt to frame your argument from this perspective. I have raised many issues, and if you are willing to seriously rebut them, there is much that can be done to enhance your experiments. The writing flaws will be difficult to address in the short term, but I hope you take them seriously. Good luck.

**Questions:**

See the weakness.

---

> ### Author Response · Authors · 2024-11-21
> **Official Comment by Authors (1)**
>
> Thanks for your constructive and helpful comments!
>
> **W1. Attempt on a classic benchmark like WebShop**
>
> Thanks for your reminder!
>
> We are sorry that our claim about the reason to choose web navigation is not clear enough. The reason why we choose web navigation tasks is that **it is complex enough**. However, the reason why we do not choose OS tasks is that the environment is hard to set up and less accessible to academia compared with web navigation tasks. (benchmarks like OSWorld [1] need a bunch of servers with virtual machines to run and were also **unstable by the time we did this work**). We will update this and explain it more clearly in the intro part of our final version to eliminate confusion.
>
> The reason why we choose WebArena and VisualWebArena, as stated at the beginning of our experiments section, is that they are more challenging and also more diverse and realistic in terms of task type. WebShop focuses mainly on shopping tasks and is simpler in terms of action space (only 2 types of actions, click and search) and observation space (simple web pages).
>
> However, we understand your concerns. To make our results more persuasive, we evaluated CER on WebShop with 500 tasks following the ReAct setup. Since CER works in a streaming setting **without any retry** (trial number = 1), we reproduce ReAct and RAP results for comparison since they can be easily applied when only one trial is allowed. Methods like Reflexion or LATS either require multiple trials or a lot of sampling steps, which is not suitable for our comparison considering fairness. Each case is only attempted once, and CER runs online. We use GPT-4o-2024-05-13 as in the paper. The results are as follows:
>
> | Method | SR |
> | --- | --- |
> | ReAct | 30.0 |
> | RAP | 33.8 |
> | ReAct w/ $\text{CER}_\text{online}$ | 34.8 |
>
> For the interpretation of the results, it should be noted that:
>
> 1. **WebShop cannot fully show the advantage of CER** because the task type is almost the same across the dataset (3-step shopping task). So RAP can benefit from it more because it directly uses past trajectory. **CER specializes in distilling fine-grained skills and environment dynamics, which are especially useful for complex environments** (e.g., many diverse and complex webpages) and long-horizon tasks (i.e., tasks that take many steps).
> 2. In our experiment, RAP has access to ground truth reward, while CER does not. This is because RAP works only for successful trajectories.
> 3. For more differences in terms of methodology between RAP and CER, you may refer to W4.
>
> **W2. Abuse of terms and lack of theoretical framework analysis**
>
> Thanks for pointing out the inspiring work! (https://openreview.net/forum?id=9W6Z9IeLzc). It takes a longer time to establish such a framework. So, we will define our terms more strictly and also follow the work to add some theoretical explanations for our method in revision.
>
> **W3. Lack of experiments on open-source models**
>
> Thanks for pointing out!
>
> For reproductivity, we specify the version of the closed-source model we used, i.e., GPT-4o-2024-05-13. This endpoint should be available and stable in a relatively long term to reproduce our results. But we agree with your concerns about reproducing and also the effectiveness of weaker open-source models.
>
> We indeed encountered problems when testing CER on Llama due to the lack of computing resources. However, we try our best to test CER with the open-source model, i.e., Llama-3.1-70B, in the tight rebuttal period on the Gitlab split of WebArena (the largest website split). Due to time constraints, we directly test CER with hybrid (offline + online) learning, which is the full version of CER. We will complete experiments in other settings in the final version. Here are the results:
>
> | Method | SR |
> | --- | --- |
> | ReAct | 17.34 |
> | ReAct w/ $\text{CER}_\text{hybrid}$ | 21.94 |
>
> CER also improves the performance of Llama3.1 on WebArena by a relative improvement of 26.53%, proving its validity on weaker open-source models. We also analyzed the results and found that weaker models like llama3.1 do worse compared with GPT-4o at formatting their output when solving challenging tasks like WebArena with a larger action space. This also influences their robustness when distilling some multi-step, useful, and well-formatted skills, which explains why the improvement is relatively smaller than it is on strong models like GPT-4o.
>
> (to be continued in the next comment)

---

> ### Author Response · Authors · 2024-11-21
> **Official Comment by Authors (2)**
>
> **W4. Overstate the contributions**
>
> Thank you for pointing out this problem. We have modified the contribution part in the updated paper to make it more reasonable (from lines 146 to 147): “CER distills fine-grained skills and environment dynamics from both successful and failed trajectories. Importantly, it works for offline, online, and hybrid settings.”
>
> There are indeed some related works, as mentioned in our paper’s related work part, that leverage past experiences. We will also include RAP in the final version of our paper.
>
> Still, we want to claim a few significant differences between RAP and CER here:
>
> 1. RAP utilizes **original past trajectories** to help solve the current task, while CER distills more **multi-step reusable skills and environment dynamics**. Using past trajectories directly could help most on benchmarks where tasks are highly similar and even almost the same pattern, like WebShop. However, task type is much more diverse on benchmarks like WebArena and in real-world scenarios, where CER helps more. This is because CER distills multi-step reusable workflows and environmental dynamics.
> 2. RAP only utilizes successful trajectories, while CER works for **both successful and failure trajectories**. In real-world scenarios, it is not always possible to get the ground truth reward/correctness. This is also the advantage of CER when applying it in a continual learning paradigm.
> 3. RAP works only in online settings, while CER works well **offline, online, and hybrid settings**. In our updated experiments on the full set of WebArena (as in the table in W5), offline and online settings show great synergy with each other.
>
> **W5. Marginal improvements**
>
> As listed in Table 1 and Table 2 in our paper, CER shows an absolute improvement over the baseline by 8.9% and 5.7% on WebArena and VisualWebArena, respectively. Such improvement is still significant **considering the difficulty of tasks on WebArena and VisualWebArena**. Also, for a more extensive evaluation of CER under different learning modes, we also supplement the experiments for CER with offline and hybrid settings on the full set of WebArena.
>
> | Method | Average SR (%) | Shopping | Shopping admin | Reddit | Gitlab | Map |
> | --- | --- | --- | --- | --- | --- | --- |
> | BrowserGym | 24.26 | 26.56 | 28.02 | 22.80 | 21.43 | 18.35 |
> | AutoEval | 20.20 | 25.50 | 18.10 | 25.40 | 28.60 | **31.90** |
> | $\text{CER}_\text{offline}$ | 33.42 | 29.17 | 36.81 | 33.33 | 36.73 | 29.46 |
> | $\text{CER}_\text{online}$ | 33.16 | 29.17 | 36.26 | 37.72 | 34.18 | 28.57 |
> | $\text{CER}_\text{hybrid}$ | **36.68** | **32.81** | **41.21** | **41.20** | **37.24** | 30.36 |
>
> The best-performed hybrid setting surpasses the baseline (BrowserGym) by **an absolute improvement of 12.42%,** which is significant.
>
> Lastly, **we are very grateful to get such detailed and constructive comments, which will help us improve the work further**. In terms of writing problems, we would polish it and frame our work better in revision. I hope our replies help. Please feel free to follow up if you have any further questions!

---

> > ### Comment · Reviewer_FBmG · 2024-11-24
> > **Best Luck With Your Paper**
> >
> > Sorry to reply to your rebuttal so late. You definitely addressed some of the concerns I've mentioned before and there are some things I want to share:
> >
> > For the OS bench benchmark, you should definitely try out AgentBench, which consists of the easiest-to-configure OS bench and DB bench. It is much easier for you to build up your results. I know WebArena is quite complicated, so I hope you will try it out in a later revision.
> >
> > As has been shown, I appreciate your attitude and effort in the rebuttal. I will stay with my score and wish you all the best. Try to make your paper strict and scientific from the beginning next time. Best of luck.

---

### Official Review · Reviewer_Q6jT · 2024-11-02

**Soundness:** 3
**Presentation:** 4
**Contribution:** 3
**Rating:** 6
**Confidence:** 4

**Summary:**

This paper presents Contextual Experience Replay (CER), a training-free framework to enable efficient continual learning for language agents in sequential decision-making tasks. Specifically, the methodology includes three processes: distilling experiences from trajectories, retrieving experiences from the buffer, and decision-making with contextual experience replay. The methodology can be divided into offline, online and hybrid versions. Experiments conducted on WebArena and VisualWebArena demonstrate the effectiveness of the approach.

**Strengths:**

1. Continual learning for language agents is a very important topic, and this paper focuses on it by choosing the Web navigation task for study.
2. This paper is well-crafted, with the authors' perspectives clearly and effectively conveyed.
3. The experiments in the paper are thorough, and the authors provide a comprehensive and convincing analysis.

**Weaknesses:**

1. This paper lacks an evaluation of failed cases and does not address common reasons for the agent’s failure in performing web navigation tasks. For example, failures could arise from misleading input experiences or limitations in the model’s grounding capabilities. To strengthen the evaluation, the authors could consider conducting a manual error analysis to understand the bottleneck for continual learning on web navigation tasks.
2. Additionally, the paper does not explore whether tasks previously failed by the agent can be successfully completed upon reattempt, following a process of summarizing and learning from those failures. To validate this, the authors could include an ablation study that retests previously failed tasks to observe whether the agent’s capabilities have improved.

**Questions:**

1.  **Token Efficiency of CER**: The paper mentions that CER can operate with fewer toknes compared to other methods. Could the authors provide statistics on token usage? Additionally, when reporting token efficiency, is the token expenditure from processes like distillation included?

2. **Fairness of the "Online Setting" in the Main Experiment**: I am somewhat confused by the authors' claim that the "online setting" is fair in the main experiment. In my view, the agent should not use experiences gained from the test set during the testing phase, as this could bias the results in its favor. If the agent leverages data from the test set more than other methods do, this could inherently be unfair. Perhaps dividing the benchmark into separate training and testing sets would be a more balanced approach. I recommend the authors to:

  - Clarify how their online setting compares to standard evaluation practices in continual learning.
  - Discuss potential biases introduced by this approach and how they might affect the interpretation of results.
  - Consider running an additional experiment as suggested if the potential biases are significant.

3. **Clarification on Experience Retrieval Using LLMs (Section 3.2)**: Section 3.2 lacks clarity on how experiences are retrieved. Are large language models specifically employed to select the most similar experiences? If so, how does this retrieval approach compare to traditional algorithms in terms of advantages and disadvantages? Furthermore, does the context window limitation of LLMs affect retrieval effectiveness? Lastly, what is the accuracy of this retrieval method? It would be beneficial to include any metrics or evaluations used to assess retrieval performance.

4. **Further Explanation and Examples for "Environment Dynamics"**: It would be helpful if the authors could provide a few more examples to clarify the definition of "environment dynamics". Additionally, how transferable are the experiences obtained from certain websites to other, potentially new, websites? I hope the authors can give me some examples from experiments to explain this.

---

> ### Author Response · Authors · 2024-11-21
> **Official Comment on Weaknesses by Authors**
>
> **W1. Error analysis of failure cases**
>
> Thanks for your advice! We manually analyze the failure cases and summarize some common and major error reasons as follows. These will also be added to our final version:
>
> 1. **Inability to backtrack and explore alternative paths**: the agent can hardly realize that it needs to backtrack and try another branching path when reaching a bad state. It just gets stuck around the bad state and can hardly get out of it. This makes it hard for the agent to recover from a wrong action. This might be solved through exploration methods like tree search.
> 2. **Limited utilization of trajectory history**: although the agent is provided with past trajectory history, it can hardly make use of it or understand it well. For example, for a task where the agent is asked to count the items the user bought in a few orders, although the thinking steps are recorded in history where the numbers of items in each order are, it is even hard for it to know the orders it has navigated to or the number of items in the navigated orders.
> 3. **Failure to verify relevant context before action**: For cases where the agent is asked to find “the count of comments that have received more downvotes than upvotes for the user who made the latest post on the books forum” after the agent goes to the forum page, it just clicks the first post without considering whether it is the latest one or realizes that it needs to sort the forums first.
>
> We also found that **misleading bad experiences is not a big problem**, according to Section 5.5 of our paper. The results in Table 6 show that the gap between distilling experiences from successful trajectories only or from all trajectories is acceptable, and the experiences from failed cases are not a bottleneck for CER.
>
> Firstly, the number of misleading experiences is small. This is because the distillation module is prompted to check the completeness of a subtask from its trajectory before it is distilled into a subtask. Additionally, when we investigated the results, we found that the agent can still make its own correct decision in many cases, even when provided with bad experiences.
>
> **W2. Reattempt of failure cases**
>
> Thanks for pointing out! We did not include this in our experiment because we thought this setting did not fit real-world scenarios before. However, it is indeed a good probe to investigate whether the agent improves itself after learning from a few tasks.
> So we carried out the experiment on the shopping admin split of WebArena. We found that the agent can successfully complete many previous failure cases, leading to a performance improvement **from 36.3% to 42.3%** as in the table.
>
> | Method | SR (%) | $\Delta$ |
> | --- | --- | --- |
> | CER | 36.3 | 0 |
> | CER w/ reattempt | 42.3 | 16.5% |
>
>  We will include the results on the full set later in our final version.

---

> ### Author Response · Authors · 2024-11-21
> **Official Comment on Questions by Authors**
>
> **Q1. Token efficiency**
>
> We calculated the token costs of our method. The results are as follows:
>
> | Method | Success rate | Additional input tokens / task | Additional costs percentage | Relative performance improvements |
> | --- | --- | --- | --- | --- |
> | BrowserGym | 24.26 | 0 | 0% | 0% |
> | $\text{CER}_\text{offline}$ | 33.42 | 7885 | 5.8% | 37.76% |
> | $\text{CER}_\text{online}$ | 33.16 | 16337 | 11.2% | 36.69% |
> | $\text{CER}_\text{hybrid}$ | **36.68** | 20631 | 17.3% | 52.50% |
>
> (Note: additional input tokens here mean the extra tokens CER takes besides the inference cost of the base agent method. The inference cost of CER is roughly the same as the baseline because CER is built on top of it.)
>
> For cost comparison with the Tree Search method on VisualWebArena. The original paper does not provide their token costs, and it is costly to reproduce their results. So, we conceptually compare their method with ours in the paper. Here we show that for CER, the additional cost is less than 20% w.r.t the inference cost. In our case, the inference cost is the baseline agent with a max step of 30. According to their algorithm, the token cost of the tree search method is comparable to a baseline agent with a max step of 100. So CER is much more efficient than that (**36:100**). It should be noted that CER can also be built on top of the tree search method.
>
> **Q2. (Un)fairness of the “Online setting”**
>
> Thanks for pointing out! We provide more detailed explanation for this question in the **general response part**. Please feel free to raise questions if any aspects we do not include.
>
> **Q3. Clarification on retrieval using LLMs**
>
> In our framework, we use chain-of-thought prompting to prompt the LLM to choose **the most useful experiences for the task rather than simply the most similar experiences**.
>
> 1. Comparison between LLM and traditional retrieval method:
>
>     We run experiments on the Gitlab subset of WebArena and get the following results. For embedding-based retrieval, we use Instructor Embedding (Su et al., 2023) here since it can generate customized embedding by different aims, and we use the task goal as the query in our case:
>
>     | Method | SR |
>     | --- | --- |
>     | CER w/ embedding-based retrieval | 32.65 |
>     | CER w/ LLM-based retrieval | **34.18** |
>
>     As in the table, LLM does a better job for this type of retrieval. Specifically, here is the comparison:
>
>     - Advantage: LLM with CoT prompting can decompose the task goal and choose the most helpful skill accordingly. However, traditional retrieval methods can hardly capture the relationship between the skills and the task when the task becomes complex.
>     - Disadvantage: Retrieval with LLM can be more expensive compared to the traditional algorithm, especially when the number of tasks is large.
>
> 2. Context window problem of LLM:
>
>     We do not meet such a problem in our experiments. The results above also show that LLM is still a better choice for retrieval.
>
>     Each experience is around 50 tokens in our implementation. So the total length of all existing experiences is still acceptable. So, we did not find evident problems when the number of skills increased. It could be a problem if the number of experiences becomes huge as the accumulation of experience buffers. Strategies like hierarchical selection can be applied to improve the retrieval performance if such problems arise.
>
> 3. Accuracy of retrieval
>
>     It is hard to measure the accuracy of skills retrieved in our scenario, i.e., web agent tasks, since the correctness is hard to judge and it can be time-consuming to annotate by humans. So we run experiments on the Gitlab split with LLM-based retrieval and embedding-based retrieval, hoping to **reflect the effectiveness of these two retrieval methods from the final performance**. As in the table above, CER with embedding-based retrieval performs worse than LLM-based retrieval, indicating that it could have worse accuracy than LLM-based retrieval.
>
>
> **Q4. Explanation of environment dynamics**
>
> Here is an example of environment dynamics (after stringifying) in Fig 2 in our paper; this is from one of the cases on Reddit split of WebArena:
>
> ```
> Dynamic 1: Forums page;
>
> Forums page contents: it shows a list of different forums;
>
> Potential usages: navigate to different forums;
>
> URL: http://xxxx:9999/forums
> ```
>
> Such dynamics can:
> 1. Help the agent be aware of some useful web pages. This will influence their decision-making implicitly.
> 2. As an important characteristic of web navigation, the agent can directly navigate to the page with the URL, which is effective and reduces the number of actions.
>
> As a website-specific knowledge, It could be hard to transfer such dynamics to other websites. However, it can be acquired after some attempts on the website and improve overall performance, as our ablation study indicates in Section 5.6.
>
> ---

---

### Official Review · Reviewer_uVLQ · 2024-11-03

**Soundness:** 2
**Presentation:** 2
**Contribution:** 2
**Rating:** 3
**Confidence:** 4

**Summary:**

This paper proposes CER, which distills foundation model agent experience into environment dynamics knowledge and action guidance to help with decision making, in online or/and offline setting. The authors presented performance improvements on realistic web navigation domains WebArena and VisualWebArena, demonstrating the effectiveness of the proposed method.

**Strengths:**

- The authors conducted a variety of experiments to analyze the performance from different perspectives.
- CER shows some promise to learn from failed experience of foundation model agents.

**Weaknesses:**

- Missing important baselines and overstating paper contribution: The authors claim that the method is the first framework that distills experiences into both environment dynamics and decision-making patterns from both successful and failed trajectories. However, there are existing works that do the same and also work on challenging tasks like WebArena. For example,
  - AutoGuide[1] summarizes contextual guidelines from successful and failed trajectories, where the state serve as a similar role as the dynamics and the guideline serves at the same role as decision making patterns.
  - AutoManual[2] autonomously interact with the environment and categorizes environmental knowledge into rules in an online manner.
- The writing of the method section can be further improved. The author claim that a major contribution of the method is to learn from both successful and failed experience. However, it is unclear how this is done until the experiment section, making it a bit confusing.

**Questions:**

- Is the current online experiment conducted in the order of the default task ID of the benchmarks? Are the any specific order that will further favor the online learning? For example, a curriculum learning like easy to hard ranking of the tasks.
- How is the hyper-parameter top-k selected? How will a different k influence the performance?


---
[1] Fu, Yao, Dong-Ki Kim, Jaekyeom Kim, Sungryull Sohn, Lajanugen Logeswaran, Kyunghoon Bae, and Honglak Lee. "Autoguide: Automated generation and selection of state-aware guidelines for large language model agents." arXiv preprint arXiv:2403.08978 (2024).

[2] Chen, Minghao, Yihang Li, Yanting Yang, Shiyu Yu, Binbin Lin, and Xiaofei He. "AutoManual: Generating Instruction Manuals by LLM Agents via Interactive Environmental Learning." arXiv preprint arXiv:2405.16247 (2024).

---

> ### Author Response · Authors · 2024-11-21
> **Official Comment on Weaknesses by Authors**
>
> Thank you for your constructive comments!
>
> **W1. Missing baselines and overstating contribution**
>
> We appreciate your perspective.
>
> For the claim of contribution, we have modified it to make it more reasonable in our updated paper (from lines 146 to 147): “CER distills fine-grained skills and environment dynamics from both successful and failed trajectories. Importantly, it works for offline, online, and hybrid settings.”
>
> For the two related works, we are sorry to miss them in our paper. There are some significant differences between CER and these two related works. We will compare them one by one here and will add the comparison to our final version.
>
> **For AutoGuide**, they only report results on Reddit split of WebArena.
>
> Here are the results of CER and AutoGuide on Reddit split:
>
> | Method | Success Rate (%) |
> | --- | --- |
> | $\text{CER}_\text{offline}$ | 33.3 |
> | $\text{CER}_\text{online}$ | 37.7 |
> | $\text{CER}_\text{hybrid}$ | 41.2 |
> | AutoGuide | 43.7 |
>
> Some important comments on the result:
>
> 1. AutoGuide’s setup is a bit different and **could be unfair to compare** with us because they use human demonstrations of tasks from the test set to extract the guidelines.
> 2. Their results on other splits are somehow **hard to reproduce** because of the similar reason - the ground truth trajectories are hard to collect.
>
> In terms of methodology:
>
> 1. AutoGuide uses **ground truth** trajectories of tasks in **test set** for learning while CER does not. CER can learn from a larger range of trajectories, including successful and unsuccessful ones.
> 2. AutoGuide summarizes single-step guidelines only, and CER distills **skills** (multi-step reusable workflows) and **environment dynamics** (important state information).
> 3. AutoGuide only works for offline data, while CER works for **offline, online, and hybrid settings**. These settings are divided by the data source, from offline collected data, online past trajectories, or both. CER shows its potential under all these settings. We also show the synergy between offline and online learning for CER. Results can be found in the table in General Response.
>
> **For AutoManual**, we reproduce their method on the Gitlab split of WebArena. Here are the results:
>
> | Method | Success Rate | Input tokens / task |
> | --- | --- | --- |
> | AutoManual | 25.51 | 465910 |
> | $\text{CER}_\text{offline}$ | 36.73 | 143442 |
> | $\text{CER}_\text{online}$ | 34.18 | 161908 |
> | $\text{CER}_\text{hybrid}$ | **37.24** | 139571 |
> 1. In terms of results, CER outperforms AutoManual by mostly 11.73%. Also, CER takes only about ⅓ tokens of AutoManual.
> 2. In terms of methodology and setups, here are the differences
>     - AutoManual runs on the test set twice, while CER runs only once. This makes CER more realistic and easy to deploy in the real world.
>     - AutoManual runs only in the online setting, while CER works well for offline, online, and hybrid settings. CER also shows a good **synergy between offline and online learning**.
>     - CER also includes environment dynamics (important state information) when distilling the experiences, while AutoManual does not.
>
> Note: They also only report scores on Reddit. The reasons why we reproduce the method on Gitlab split are:
>
> - It is costly and also time-consuming (as indicated in the table, it takes multiple times the costs) to reproduce their method on the full set. Gitlab is the largest split among all website splits (about ¼ of the full set).
> - They use 2 human demonstration examples from the test set of Reddit split for few-shot prompting. So it is more fair to test on another website split.
>
> **W2. Writing of method part**
>
> Thanks for your advice. We will improve the method part to make it clearer in our final version. Please let us know if you have any specific parts to clarify.

---

> ### Author Response · Authors · 2024-11-21
> **Official Comment on Questions by Authors**
>
> **Q1. Influence of order**
>
> Yes, the current online experiment is conducted in the default order.
>
> It is interesting to investigate the influence of task order. Since only VisualWebArena provides the difficulties of tasks, we ran the experiment on the classifieds split of VisualWebArena to investigate the effectiveness of the curriculum learning style. The results are as follows:
>
> | Task Order | SR |
> | --- | --- |
> | Original | 27.0 |
> | Easy-to-hard | **28.1** |
>
> As indicated in the table, curriculum learning improves the performance of CER. This is intuitive because the agent can learn skills easily from easy tasks and apply them to harder tasks. However, it should be noted that this can hardly be applied in real-world situations because it is hard to measure the difficulty of tasks beforehand.
>
> **Q2. Choice and influence of k**
>
> We chose k according to the difficulty of the tasks to be solved because harder tasks will need more steps to solve, thus potentially more skills needed. Tasks on WebArena usually can be decomposed into less than 5 subtasks so we choose k=5. To empirically investigate the influence, we compare the performance of different choices of k on the Gitlab split of WebArena.
>
> | k | SR |
> | --- | --- |
> | 1 | 23.98 |
> | 3 | **34.69** |
> | 5 | 33.16 |
> | 7 | 32.65 |
>
> Results show that if k is too small, it will negatively influence the performance, while larger k will not influence the performance too much. This is because small k, e.g., k=1, might lead to the omission of some necessary experiences. We recommend **choosing k according to the difficulty of the problems**, i.e., consider how many distinct subtasks the original task can be split into. This approximately represents the largest number of useful skills needed to complete the task. Also, selecting an appropriate value for k is not a big problem generally because, as our results indicate, **performance is not highly sensitive to slight changes in k**.

---

> ### Comment · Reviewer_uVLQ · 2024-11-24
>
> I appreciate the effort the authors have put into the rebuttal. My responses are as follows:
>
> **Points of Disagreement**:
>
> > AutoGuide summarizes single-step guidelines only, and CER distills skills (multi-step reusable workflows) and environment dynamics (important state information).
>
> According to their prompt in the paper, AutoGuide summarizes guidelines at the state level because it says: *Please refer to 'the previous actions' and 'Demonstration actions in later steps' to generate more accurate descriptions of your purpose and the sequence of actions to achieve the purpose.* And as I mentioned, the state summarization is also very similar to the *important state information* you refer to.
>
> > CER also includes environment dynamics (important state information) when distilling the experiences, while AutoManual does not.
>
> However, AutoGuide also incorporates this aspect.
>
> **Points regarding contribution**:
>
> > AutoGuide only works for offline data, while CER works for offline, online, and hybrid settings.
>
> > AutoManual runs only in the online setting, while CER works well for offline, online, and hybrid settings.
>
> AutoGuide corresponds to the offline setting, and one of AutoManual’s contribution is its ability to collect experience in an online rule system. The statement that CER “shows a good synergy between offline and online learning” does not strike me as a significant contribution.
>
> > AutoGuide uses ground truth trajectories of tasks in test set for learning while CER does not.
>
> First of all, according to the paper, AutoGuide splits the original Reddit tasks into a training set and a test set, ensuring that these sets are non-overlapping. Additionally, AutoGuide’s method only requires successful and failed trajectories as offline data, with the use of human demonstrations being a design choice for collecting successful trajectories. On the other hand, CER collects successful and failed trajectories automatically through the agent itself, but without employing any special exploration strategies. Therefore, the distinction seems to be that CER demonstrates it is feasible for an agent to collect successful trajectories online within these domains. Personally, I do not find this contribution substantial enough. However, I am open to further discussion if the authors, other reviewers, or the AC have a different perspective on this matter.
>
> **Regarding the experiments**:
> > curriculum learning improves the performance of CER
>
> > choice and influence of k
>
> Thank you for the new experiments, it will be useful to see them in a later version of the paper.
>
> To conclude, my major concern remains the contribution of this paper, as it appears very similar to previous works. Therefore, I will retain my score.

---

> ### Author Response · Authors · 2024-11-29
> **Clarification of our contribution**
>
> Thanks for pointing out our mistake — you’re right that AutoGuide also summarizes over multiple steps. However, we believe CER still has a key distinction from previous methods:
>
> **Need for success/failure labeling**
>
> For AutoGuide, it needs a pair of successful and failed trajectories from the same task to compare and summarize, while CER learns from any unlabeled trajectories, making it significantly more versatile and broadly applicable. Importantly, AutoGuide needs both success and failure labeling. As mentioned in their paper:
> > Among the 19 human demonstrations provided in WebArena, **17 of them** successfully completed the tasks. We directly run ReAct on **the successful tasks** to collect **failure actions** and generate guidelines correspondingly
> >
> Both success and failure evaluations here rely on ground-truth evaluators provided by WebArena. Although benchmarks like WebArena provide ground-truth evaluators, **the evaluation of a trajectory is hard and time-consuming in the real world**. Evaluation from a human is time-consuming, and evaluation from an LLM is hardly reliable. So it is important to investigate how to learn without access to such evaluation.
>
> For AutoManual. It also needs labeled trajectories as AutoGuide and there are several other distinctions and drawbacks compared with CER, as mentioned in our last reply.
>
> Imagine a real-world scenario: AutoGuide is an agent that requires many demonstrations and also many manual evaluations to check the correctness of its attempts from you; AutoManual needs to try all tasks you want it to complete first and requires you to check the correctness one by one before it starts working (it also has multiple times costs); CER is an agent that does not require you to do anything and can improve itself automatically and continuously. Which one do you prefer?
>
> **Reclarification of environment dynamics**
>
> In our previous reply, we clarified that CER uniquely incorporates environment dynamics, a feature neither AutoGuide nor AutoManual possesses. To make it clear, environment dynamics in our method provide **an extra shortcut to navigate to a specific webpage with its URL**, improving efficiency and reducing the number of actions required.
>
>
> For example, as shown in Fig. 2 of our paper:
>
> “ Dynamic 1: Forums page;
>
> Forums page contents: it shows a list of different forums;
>
> Potential usages: navigate to different forums;
>
> URL: http://hostname:9999/forums ”
>
> The agent can directly go to the Forums page with its URL. This feature is analogous to using browser history to revisit familiar pages, allowing the agent to navigate websites more efficiently. This is not included in AutoGuide or AutoManual. We will also add such an explanation in the revision.

---

> ### Comment · Reviewer_uVLQ · 2024-12-01
>
> Thank you for the response.
> > In our previous reply, we clarified that CER uniquely incorporates environment dynamics, a feature neither AutoGuide nor AutoManual possesses. To make it clear, environment dynamics in our method provide an extra shortcut to navigate to a specific webpage with its URL, improving efficiency and reducing the number of actions required.
>
> Regarding the example you provided, the first part (Dynamic 1: Forums page; Forums page contents: it shows a list of different forums) appears to be similar to AutoGuide's state summarization approach. Furthermore, utilizing URLs is a widely adopted technique in current web navigation methods. For instance, in [1], the authors explicitly record URLs as part of the state representation for navigation purposes.
>
> > Imagine a real-world scenario: AutoGuide is an agent that requires many demonstrations and also many manual evaluations to check the correctness of its attempts from you; AutoManual needs to try all tasks you want it to complete first and requires you to check the correctness one by one before it starts working (it also has multiple times costs); CER is an agent that does not require you to do anything and can improve itself automatically and continuously. Which one do you prefer?
>
> Thank you for pointing this out. Regarding access to a ground-truth evaluator, I do not think this as inherently disadvantageous. As evidenced in your Table 6, access to such an evaluator appears to lead to improved outcomes. Also, the extent of the performance gap in other domains remains unclear.
>
> However, I do think the authors make a valid point by demonstrating that CER eliminates the need for a ground-truth evaluator within the WebArena domain. Nonetheless, this distinction is not adequately emphasized in the paper. If the authors consider this the primary difference, I believe the paper may require significant revisions to highlight this contribution. Given that method-wise, the methodology is highly similar to prior works, this contribution of **showing the efficacy of such methods in the absence of a ground-truth evaluator** should be emphasized, instead of focusing on distilling and retrieving dynamics, skills, with online or/and offline data, which are all mentioned in previous works we discussed and the way of writing can be misleading.
>
> ---
> [1] Wang, H., Li, T., Deng, Z., Roth, D., & Li, Y. (2024). Devil's Advocate: Anticipatory Reflection for LLM Agents. arXiv preprint arXiv:2405.16334.

---

### Official Review · Reviewer_VqHM · 2024-11-04

**Soundness:** 2
**Presentation:** 2
**Contribution:** 2
**Rating:** 5
**Confidence:** 4

**Summary:**

This submission proposes Contextual Experience Replay (CER) as a continual learning approach that accumulates experiences from the task domain and retrieves relevant ones for later tasks, for building web navigation agents. Specifically, upon obtaining a new experience, they distill it into dynamics-related and behavior-related descriptions, which can be retrieved for augmenting the LLM agent's decision-making on different but relevant or similar tasks. The authors perform an empirical evaluation of their approach on WebArena and VisualWebArena tasks and demonstrate that this approach outperforms the compared baselines. They also provide additional empirical analyses in multiple aspects, such as continual learning concerns (knowledge preservation vs acquisition) and different types of source data.

**Strengths:**

- The proposed approach is overall sound. Pre-trained LLMs often lack domain knowledge due to the required specificity and continuous updates (like websites), and distilling knowledge from experiences for incorporation into LLMs' input contexts can be useful to mitigate such issues.
- The authors provide multiple empirical analyses with the proposed approach, such as the source of trajectory data and knowledge preservation and acquisition study. These can provide a deeper understanding of the proposed approach.
- Overall, the manuscript is easy to follow, especially with the well-visualizing figures.

**Weaknesses:**

- While the continual accumulation of experiences are often useful in practice and the accumulated knowledge can be useful information, doing so during the evaluation on the test task set from each of those benchmarks can make the comparison with other baseline approaches unfair. In most benchmark setups, gaining knowledge about other test tasks usually provides noticeable performance boosts primarily due to the task similarities and increased data size. To properly compare the performance of the proposed approach with the baselines in a "held-out" setting, the "self-guided explorations" configuration from Table 3 seems more appropriate, as it performs the same but without the knowledge of what will happen during the completion of each task.
- I notice that there is a methodologically close paper that is not cited and compared: Fu et al., 2024 (AutoGuide: Automated Generation and Selection of State-Aware Guidelines for Large Language Model Agents). For instance, I think there is some correspondence as follows: (CER's distillation, their guideline extraction), (CER's top-k dynamics and skill retrieval, their top-k guideline retrieval).
- There are some minor typos in the writing, such as missing periods and mis-capitalizations.

**Questions:**

Please check out the weaknesses section.

---

> ### Author Response · Authors · 2024-11-21
> **Official Comment by Authors**
>
> Thank you for finding our approach sound and recognizing our empirical analysis!
>
> **W1. (Un)fairness of the online setting**
>
> Thank you for pointing out! Since this is also mentioned by another reviewer, we answer it in the first part of the general response. Please also feel free to raise if you have any further questions.
>
> **W2: Comparison with AutoGuide**
>
> Here are some significant differences between AutoGuide and CER:
>
> - AutoGuide uses **ground truth trajectories** of tasks in the **test set** for learning, while CER does not. CER can learn from a larger range of trajectories, including successful and unsuccessful ones.
> - AutoGuide summarizes single-step guidelines only, and CER distills skills (multi-step reusable workflows) and environment dynamics (important state information).
> - AutoGuide only works for offline data, while CER works for **offline, online, and hybrid settings**. These settings are divided by the data source, from offline collected data, online past trajectories, or both. CER shows its potential under all these settings. We also show the synergy between offline and online learning for CER. Results can be found in the table in General Response.
>
> We will cite and add the comparison in our final version.
>
> **W3: Writing problem**
>
> Thank you for reminding us of the typos! We have already corrected them in our paper. We will also double-check them in our final version.

---

> > ### Comment · Reviewer_VqHM · 2024-12-03
> > **Response to Authors**
> >
> > Thank you for providing the detailed response regarding W1. While the provided response do add values, I still have remaining concerns as follows. I think employing *human annotations* as offline data source makes it similar to existing work, where proper continual learning with its own experience would require a different setup.
> >
> > Regarding W2, according to the paper, they use successful and unsuccessful trajectories but without test set tasks. I am not suggesting there are no differences, but given the comparable components between these two papers, more discussions are needed in the paper.
> >
> > Overall, while I appreciate the authors' effort in providing the rebuttal, my perspective on this work remains similar.

---

### Author Response · Authors · 2024-11-21
**General Response**

We sincerely thank all the reviewers for their feedback and constructive comments. Here are responses to some general questions.

Note: offline and online are defined by the data source as in our paper. The offline setting is to learn from some offline training data, i.e., trajectories, the online setting is to learn from past trajectories during inference time, and the offline + online (hybrid) setting has both.

1. (Un)fairness of the online setting
    - For the baseline, the two main baselines we compared with are BrowserGym (Drouin et al., 2024) and AutoEval (Pan et al., 2024). We think **the comparison is fair between $\text{CER}$ and them.**
        - BrowserGym only changes the observation space and action space (environment-level). $\text{CER}$ is built on top of their environment, so our improvements should be considered fairly.
        - AutoEval even has multiple retries on the same task on the test set, so it has a similar “bias”. So it is fair to directly compare performances between $\text{CER}$ and it.

        So the overall improvements can be fairly interpreted.

    - **Offline setting also works well**. We ran additional experiments in an offline setting and offline + online setting on the **full set** of WebArena. For the offline setting, we use a few human demonstrations (less than 10 tasks per website, no overlapping with the test set) as the initial trajectories to distill from. Here are our results: as in the table, $\text{CER}\_\text{offline}$ shows comparable and even better results than $\text{CER}\_\text{online}$. **It proves that $\text{CER}$ can still work well without learning from the test set.** It also shows that with the “warm up” of offline learning, $\text{CER}$ can perform even better under offline + online settings.
        | Method | Average SR (%) | Shopping | Shopping admin | Reddit | Gitlab | Map |
        | --- | --- | --- | --- | --- | --- | --- |
        | BrowserGym | 24.26 | 26.56 | 28.02 | 22.80 | 21.43 | 18.35 |
        | AutoEval | 20.20 | 25.50 | 18.10 | 25.40 | 28.60 | **31.90** |
        | $\text{CER}\_\text{offline}$ | 33.42 | 29.17 | 36.81 | 33.33 | 36.73 | 29.46 |
        | $\text{CER}\_\text{online}$ | 33.16 | 29.17 | 36.26 | 37.72 | 34.18 | 28.57 |
        | $\text{CER}\_\text{hybrid}$| **36.68** | **32.81** | **41.21** | **41.20** | **37.24** | 30.36 |
    - To prove that $\text{CER}$’s improvement is not from task similarity and learning on the test set, we also introduce another metric - cross template success rate (ct-SR) in Section 5.2 of our paper. For short, this metric measures the success rate w.r.t. task templates (# of solved templates / # of all templates) which is defined in Section 5.2 of our paper. **If the improvement mostly comes from the favor of learning similar tasks on the test set, its ct-SR should be almost the same as the baseline.**

        Table 4 in the paper shows that $\text{CER}$ still improves a lot under this metric over the baseline. This indicates that the improvement **comes from solving new tasks rather than merely from the setting**.

    - The online setting itself is reasonable in some sense. Here are the reasons:
        - The only thing we have access to is the task goal in the test set. We did not use any gold trajectories or correctness of trajectories.
        - It fits real-life setups. The agent can improve itself through attempts at previous tasks, no matter whether successful or not. It also shows the framework's continual learning ability.
2. We ran additional experiments to further analyze and prove the validity of $\text{CER}$ in different settings, models, and benchmarks:
    - (Table above) $\text{CER}$ under offline, online, and hybrid settings on the **full set** of WebArena. This demonstrates that $\text{CER}$ works for different data sources, and the results also prove the synergy between offline and online data for $\text{CER}$.
    - (Reviewer FBmG) $\text{CER}$ with Llama-3.1-70B on WebArena. This shows $\text{CER}$’s validity on open-source models.
    - (Reviewer FBmG) $\text{CER}$ on WebShop. $\text{CER}$ improves the baseline by a relative improvement of 16% on WebShop, which makes the evaluation results more persuasive.
    - Analysis experiments including:
        - (Reviewer Q6jT Q3) $\text{CER}$ w/ LLM-based retrieval v.s. w/ embedding-based retrieval: LLM-based retrieval does a better job and is necessary for $\text{CER}$.
        - (Reviewer uVLQ Q1) Influence of the order of tasks: Curriculum learning indeed helps to achieve a better result, although it is not always realistic in daily scenarios.
        - (Reviewer uVLQ Q2) Different choices of k (# of experiences to retrieve): k should not be too small. It should be chosen based on the difficulty of tasks, i.e. marginally above the number of skills needed to complete the task.

---

### Meta-Review · Area_Chair_QG5S · 2024-12-28

**Metareview:**

This paper proposes CER, a framework for web navigation agents to leverage past experiences for self-improvement. The authors evaluate CER on two web navigation tasks and demonstrate its effectiveness.

The common concerns raised by the reviewers lies in several aspects:

1, Writing style: The paper uses many terms without sufficient explanation with overclaim.

2, Missing important baselines: there are several existing methods proposed to exploit the past experiences in LLMs. Although these methods use past experiences in different ways, it is important to includes the methods with detailed discussion.


The authors provide additional experiments, which partially addresses the reviewers' concerns. However, the aforementioned still remains. I suggest the authors to consider the reviewers' comments to improve the draft.

**Additional Comments On Reviewer Discussion:**

The authors provides additional experiments to strengthen the paper. However, the common concern raised by the reviewers, i.e., the writing style and the overclaim about novelty without citing the important baselines, has not been addressed.

---

### Decision · Program_Chairs · 2025-01-22

Reject